# Unique Gene Expression Profiles within South Africa Are Associated with Varied Chemotherapeutic Responses in Conventional Osteosarcoma

**DOI:** 10.3390/cancers16183240

**Published:** 2024-09-23

**Authors:** Phakamani G. Mthethwa, Thilona Arumugam, Veron Ramsuran, Anmol Gokul, Reitze Rodseth, Leonard Marais

**Affiliations:** 1Dr Pixley Ka Isaka Seme Memorial Hospital, University of KwaZulu-Natal, Nelson Mandela School of Clinical Medicine, 310 Bhejane Street, KwaMashu, Durban 4360, South Africa; 2Centre for the AIDS Programme of Research in South Africa (CAPRISA), School of Laboratory Medicine and Medical Sciences, College of Health Sciences, University of KwaZulu-Natal, Durban 4041, South Africa; arumagamt@ukzn.ac.za (T.A.); ramsuranv@ukzn.ac.za (V.R.); anmolgokul@gmail.com (A.G.); 3Nelson Mandela School of Clinical Medicine, College of Health Sciences, University of KwaZulu-Natal, 719 Umbilo Road, Durban 4001, South Africa; rodseth.durand@gmail.com (R.R.);

**Keywords:** osteosarcoma, gene expression profiles, chemotherapy response

## Abstract

**Simple Summary:**

This study aimed to determine the gene expression profiles associated with chemotherapeutic responses in conventional osteosarcomas (COS) within South Africa. We observed a significant downregulation in the *ATP binding cassette subfamily C members* (*ABCC3* and *ABCB1-p-glycoprotein*), *excision repair cross-complimenting group 1* (*ERCC 1*), *replication factor C subunit 1* (*RFC1*), and *tumour protein 53* (*p53*) genes in the COS tumours compared to the healthy donors. Furthermore, an upregulated *ERCC1* gene expression level predicted a poor chemotherapeutic response. Additionally, the predictors of COS chemotherapeutic response comprised age, chondroblastic and osteoblastic histological subtypes, and *ABCC3*, *ERCC1*, and *RFC1* gene expression.

**Abstract:**

Background: We determined the predictive gene expression profiles associated with chemo-response in conventional osteosarcomas (COS) within South Africa. Materials and methods: In 28 patients, we performed an RNA extraction, cDNA synthesis, and quantitative analysis using the RT-PCR 2^−∆∆CT^ method to determine the fold change in gene expression alongside GAPDH (housekeeping gene). Results: We observed a significant downregulation in the mRNA expression profiles of *ABCB1-p-glycoprotein* (*p* = 0.0007), *ABCC3* (*p* = 0.002), *ERCC1* (*p* = 0.007), *p-53* (*p* = 0.007), and *RFC1* (*p* = 0.003) in the COS patients compared to the healthy donors. Furthermore, *ABCB1-p-glycoprotein* (*p* = 0.008) and *ABCC3* (*p* = 0.020) exhibited a significant downregulation in the COS tumour tissues when compared to the healthy donors. In our univariate logistic regression, the predictors of chemotherapeutic response comprised *ERCC1* [restricted cubic spline (RCS) knot: OR −0.27; CI −0.504 to −0.032; *p* = 0.036]; osteoblastic subtype [OR −0.36; CI −0.652 to −0.092; *p* = 0.026); fibroblastic subtype [OR 0.91; CI 0.569 to 1.248; *p* < 0.001]; and mixed subtype [OR 0.53; CI 0.232 to 0.032; *p* = 0.032]. In our multivariable logistic regression, the significant predictors of chemotherapeutic response comprised age [RCS knot: OR −2.5; CI −3.616 to −1.378; *p* = 0.022]; *ABCC3* [RCS knot: OR 0.67; CI 0.407 to 0.936, *p* = 0.016]; *ERCC1* [RCS knot: OR 0.57; CI 0.235 to 0.901; *p* = 0.044]; *RFC1* [RCS knot: OR −1.04; CI −1.592 to −0.487; *p* = 0.035]; chondroblastic subtype [OR −0.83; CI −1.106 to −0.520; *p* = 0.012]; and osteoblastic subtype [OR −1.28; CI −1.664 to −0.901; *p* = 0.007]. Conclusions: In this South African cohort, we observed the unique gene expression profiles of osteosarcoma tumourigenesis and chemotherapeutic responses. These may serve as prognostication and therapeutic targets. Larger-scale research is needed on the African continent.

## 1. Introduction

Osteosarcomas are a group of aggressive primary malignant bone tumours of mesenchymal cancer stem cell origin that produce osteoid matrices. They most commonly occur in children and adolescents [1]. According to the WHO classification, central high-grade conventional osteosarcoma (COS) is the most common subtype, comprising 90% of all osteosarcoma variants [1,2]. Currently, the treatment of COS typically comprises neoadjuvant chemotherapy and wide surgical excision, followed by adjuvant chemotherapy [3,4,5,6]. The predictors of a poor survival prognosis include male sex, older age, advanced Enneking staging, non-extremity tumour, proximal long bone sites, metastasis, poor response to chemotherapy, no surgical treatment, and amputations [3,4,5,6]. The COS tumour’s chemotherapy response has emerged as the most important independent risk factor for long-term survival [3,4,5,6]. The Rosen protocol is widely utilised to measure the response to neoadjuvant chemotherapy during histological analysis after the final tumour surgical resection. A good response is defined as ≥90% tumour necrosis (Huvos grade III/IV) with a poor response defined as <90% tumour necrosis (Huvos grade I/II) [3,4].

Despite the advances in multidrug chemotherapy (MDR) and surgical procedures over the past decades, the 5-year survival of non-metastatic COS has plateaued at 60–70% [3,4,5,6]. MDR remains the most significant obstacle to improving long-term survival. The response to neoadjuvant chemotherapy is poor in up to 40–45% of cases [5]. Neither tailored postoperative chemotherapy (according to the histological response to preoperative chemotherapy agents) nor dose intensification have improved survival rates [5,6,7]. Heterogeneous gene expression patterns underpin the tumour microenvironment, with a host of molecular markers associated with chemotherapy response [8,9,10,11,12]. Examining gene expression profiles commonly known for tumourigenesis and chemoresistance in COS could assist with prognostication and guide treatment. Risk stratification using surrogate markers of disease burden and chemo-response offers great promise for the future treatment of osteosarcoma [8,9,10,11,12]. Gene therapy targeting specific genes involved in MDR is being widely researched globally but not in Africa. [8,9,10,11,12]

For instance, the *ATP Binding Cassette Subfamily C Members* (*ABCC3 and ABCB1 genes*), the latter formerly known as the *MDR1 gene or P-glycoprotein* (*P-pg*), are integral in drug uptake and transport mechanisms, thereby influencing the intracellular concentration of chemotherapy agents [9,12,13]. Contrary to drug efflux, the *Replication Factor C Subunit 1* (*RFC1 gene*), situated on the tumour cell membrane, modulates the accumulation of methotrexate (MTX), a chemotherapeutic agent commonly used in osteosarcomas [9,12,14]. On the other hand, the *Excision Repair Cross-Complimenting Group 1* (*ERCC1*) plays a pivotal role in tumour cell DNA repair mechanisms, impacting the efficacy of DNA-damaging chemotherapy agents such as cisplatin. Elevated *ERCC1* expression is associated with reduced sensitivity to cisplatin-based chemotherapy regimens [9,12,15]. Meanwhile, the tumour protein 53 (*p53 gene*) is an oncogene that plays a multifaceted role in cancer progression and chemotherapy response. Its overexpression contributes to tumourigenesis and chemoresistance by inhibiting apoptosis, a key mechanism of chemotherapy-induced cell death [9,12,16]. Exploring the expression and functional implications of conventional chemotherapy drug sensitivity genes in African populations is paramount for developing personalised treatment strategies and improving cancer outcomes in this demographic. By elucidating the interplay between genetic variations and chemotherapy response, we can pave the way for more effective and tailored cancer treatments for individuals of African descent.

This study aimed to examine the gene expression profiles associated with multidrug chemotherapy responses in central high-grade COS within the South African context, explicitly referencing the *ABCB1—p-glycoprotein*, *ABCC3*, *ERCC1*, *RFC1*, and *p53 genes.* Hopefully, the results emanating from this research will inform therapeutic targets in our population.

## 2. Materials and Methods

This prospective study comprises 28 patients diagnosed with histologically confirmed primary high-grade COS as defined by the WHO criteria [1]. We examined 28 paired (normal muscle and tumour) tissue samples from patients < 40 years of age with chemotherapy-naïve osteosarcomas of the extremities and pelvis of patients treated in out-training hospitals in South Africa between 2021 and 2022. In addition, to compare, we sampled normal muscles from nine healthy donor (non-cancerous) patients during their routine elective orthopaedic procedures. Ethical approval was obtained from the Biomedical Research Ethics Committee, and all patients or legal guardians consented to study involvement. The study proceeded once all the necessary ethical and local regulatory approvals were obtained (BREC/00002737/2021) and per the rules of the Declaration of Helsinki on human studies. The response to chemotherapy in each case was determined after neoadjuvant chemotherapy and wide surgical resection per Rosen protocol with Huvos criterion as defined previously [3,4]. We compared the gene expression profile of the primary tumours before chemotherapy induction in non-responders (NR) to neoadjuvant chemotherapy to responders (R) [8,11]. Exclusion criteria were non-conventional osteosarcoma subtypes and osteosarcoma presenting as a secondary malignancy. The clinical data of patients’ characteristics and management is provided in Table 1. Tumour volume was calculated using a previously described technique, using the formula for an ellipsoidal mass (width × height × diameter × 0.52) [17].

Immediately after CT-guided percutaneous or incisional biopsy, tumour specimens were snap-frozen in liquid nitrogen and stored at –80° C until RNA extraction. For diagnostic purposes, all samples used for RNA extraction were coded using a tissue ID corresponding to the histological section. Our previous study informed the selection of five candidate genes, which was a systematic review of the heterogeneous gene expression patterns associated with chemoresistance in COS in genome-wide sequencing studies [12]. Patients were classified according to the Huvos criteria as responders (R) if tumours exhibited ≥90% tumour necrosis and non-responders (NR) if <90% necrosis, following preoperative chemotherapy, consisting of methotrexate 8–12 g/m^2^, cisplatin 100 mg/m^2^, and doxorubicin, 25 mg/m^2^—MAP [4] [Figure 1 and Table 1].

### 2.1. RNA Extraction and cDNA Synthesis

(a)Washing: All specimens (tumour and normal tissue) contained in an Eppendorf tube were washed using 300 μL of phosphate-buffered saline (PBS). Five millimetres of tissues were placed into a PBS tube and vortexed for 60 s. The supernatant was then discarded. Then, 250 μL of PBS was added to the Eppendorf tube, and the steps were repeated. The tissues were then cut into small pieces and placed in 100 μL PBS.(b)Extraction process: We transferred the emulsified tissue and PBS into a clean 1.5 mL Eppendorf tube and added 100 μL of QuickExtract RNA Extraction Kit (LGC Biosearch Technologies, Oxford, UK). The mixture was vortexed for 2 min. The tube was then centrifuged for 2 min at 14,000 rpm. The supernatant was removed and transferred to a clean 96-well plate. We used an ND8000 nanodrop (Thermofisher, Waltham, MA, USA) to determine the concentration of RNA within our extracted samples. Thereafter, RNA was standardised to 50 ng/uL.(c)cDNA synthesis: We used the Vilo superscriPT cDNA synthesis kit (Thermofisher, USA) to synthesise cDNA. Briefly, a single reaction was made up using 4 μL of 5× VILO^TM^ Reaction Mix (Thermofisher Scientific, Carlsbad, CA, USA), 2 μL of 10× SuperScript^TM^ Enzyme Mix, and 8 μL of standardised RNA. Thereafter, the reaction mix was incubated at 25 °C for 10 min, followed by 42 °C for 60 min using the SimpliAmp Thermal Cycler (Thermofisher Scientific, Carlsbad, CA, USA). The reaction was terminated at 85 °C for 5 min. Diluted cDNA (1:400) was stored at −20 °C (Table 2).

### 2.2. RT-PCR

RT-PCR was used to determine the expression of *ABCC3*, *ABCB1*, *ERCC1*, *RFC1*, and *p53.* Briefly, a single reaction was made up using 6 μL of 5× Powerup SYBR green (Thermofisher Scientific, Carlsbad, CA, USA), 1 μL of forward primer (10 μM), 1 μL of reverse primer (10 μM), and 5 μL of cDNA (Table 3). All samples were run in triplicate per gene. The sequences for the primers used were obtained from PrimerBank (Table 3). The quality of assays was assessed by calculating the ratio of 3′:5′ features for the reference gene, GAPDH, which remained stable and when the ratio was greater than 2, the experimental results were nullified. Samples were amplified using the Quantstudio 5 (Thermofisher, Waltham, MA, USA) with the following cycling conditions: 95 °C for 2 min, 40 cycles of 95 °C for 30 s, 60 °C for 1 min, and 72 °C for 30 s (Table 3).

Relative gene expression was determined using the Livak and Schmittgen quantitation method. Gene expression of the target gene was normalised against the reference gene, *GAPDH* and it was reported as fold change using a 2-delta-delta CT method (2^−∆∆CT^) [18].

### 2.3. Outcome Measures

The primary outcome of interest was candidate gene expression levels in chemotherapy-naïve tumour specimens compared to normal muscle tissues (paired samples). Gene expression was determined by measuring the fold change in the messenger RNA (mRNA) level from a specific gene.

The secondary outcomes were the associations of mRNA expression of *ERCC1*, *FRC*, *p53*, *ABCB1*, and *ABCC3* of tumour samples with (a) the chemotherapy response as per the Huvos grading system and (b) the patient clinical factors.

### 2.4. Statistical Analysis

The statistical analysis used GraphPad Prism version 8.0.2 (263) and R (R Foundation for Statistical Computing, Vienna, Austria). Continuous variables were reported as mean (standard deviation [SD]) or median (with interquartile range [IQR]), and categorical variables as numbers and percentages unless otherwise stated. The average expression levels across individuals were compared, and the fold change for each gene was calculated. The significance level was determined using an unpaired, nonparametric, Mann–Whitney, two-tailed experimental design with a significant difference set at *p* < 0.05. We examined the relationship between gene expression and tissue histology using univariate logistic regression with gene concentrations and age modelled using a three-knot restricted cubic spline. Multivariate logistic regression was performed, and the most parsimonious model was chosen using stepwise selection by minimising the Akaike Information Criteria (AIC).

## 3. Results

### 3.1. Clinical Characteristics

The median age was 16.0 years (IQR 11.3–20.3), and there was a predilection for male sex of 64% (*n* = 18). The most common tumour locations were around the knee, involving the distal femur in 57% (*n* = 16) and the proximal tibia in 18% (*n* = 5) of cases. Other locations (*n* = 7) included the proximal femur 11% (*n* = 3), pelvis 7% (*n* = 2), proximal humerus 4% (*n* = 1), and distal radius 4% (*n* = 1). At the time of diagnosis, the median MRI tumour volume was 780 cm^3^ (IQR = 511–1133; CI = 664–1092), median alkaline phosphatase (ALP) was 237.0 U/L (IQR = 141.0–431; CI = 218.4–384.1) and lactate dehydrogenase (LDH) was 624 U/L (IQR = 229.0–663.0; CI = 386.1–801.2). Metastasis was present at diagnosis in 46.4% (*n* = 13) of the cases. The histological subtypes comprised osteoblastic (46%; *n* = 13), chondroblastic (32%; *n* = 9), mixed (11%; *n* = 3), and fibroblastic (11%; *n* = 3). Eighty-nine percent (*n* = 25) of the patients were treated with neoadjuvant chemotherapy drugs and subsequently received definitive surgery. Two patients refused treatment and sought alternative traditional medicine while one developed chemotoxicity and did not complete the treatment. All three of these patients died during follow-up. Of the remaining 25 patients, 84% (*n* = 21) did not show an adequate response to chemotherapy (Huvos criteria < 90% necrosis). The median study follow-up period was 12.7 months (IQR 9.0–17.0) [Table 4].

### 3.2. mRNA Expression of Candidate Genes

Using a Mann–Whitney test, we observed a significant downregulation in the mRNA expression levels of *ABCC3*, *ABCB1* (*p-glycoprotein*), *ERCC1*, *p53*, and *RFC1* of COS patients compared to the healthy donors as measured by RT-PCR (*p* = 0.002; *p* = 0.0007; *p* = 0.007; *p* = 0.007; and *p* = 0.003), respectively—Figure 2. Furthermore, the mRNA expression levels of *ABCC3* (*p* = 0.020) and *ABCB1*, also known as *p-glycoprotein* (*p* = 0.008), exhibited a significant downregulation in the tumour tissues of COS patients tumour when compared to the healthy donors. This is in contrast to the clear but non-significant upregulation in *ERCC1* (*p* = 0.080) and *RFC1* (*p* = 0.075) between the tumours and normal muscle tissue of the COS patients. Meanwhile, *ABCB1*, *ABCC3*, and *p53* remained non-significant (*p* = 0.205; *p* = 0.841; *p* = 0.111, respectively).

### 3.3. Chemotherapy Response

#### 3.3.1. Histology

With univariate logistic regression, the osteoblastic subtype was negatively associated with the chemotherapy response (odds ratio [OR] −0.36, CI 0.652 to −0.092; *p* = 0.026), while the fibroblastic subtype (OR 0.91, CI 0.569 to 1.248; *p* = 0.001) and mixed subtype (OR 0.53, CI 0.232 to 0.032; *p* = 0.032) were positively associated. With the addition of age and sex in a multivariate analysis, the osteoblastic histology (OR −0.45, CI −0.810 to −0.092; *p* = 0.024) remained negatively associated with the chemotherapy response, while the fibroblastic histology (OR 0.9, CI 0.523 to 1.290; *p* = 0.001) and mixed histology (OR 0.56, 0.076 to 1.046; *p* = 0.035), remained positively associated. The chondroblastic histology was not significant in the univariate (OR −0.29, CI −0.623 to 0.0351; *p* = 0.093) nor multivariate analysis (OR −0.41, −0.816 to −0.0135; *p* = 0.057).

#### 3.3.2. Gene Candidates

Using an unpaired Mann–Whitney two-tailed analysis, only the *ERCC1* gene was significantly associated with the chemotherapy response (*p* = 0.033) when comparing the non-responders against responders (Figure 3). The non-responders showed a mean 4.2-fold increase (95% CI −7.862 to –0.517) in *ERCC1* gene expression levels. In contrast, we found no significant differences in the mRNA expression levels between the responders and non-responders for *ABCC3* (*p* = 0.495), *ABCB1* (*p* = 0.231), *p53* (*p* = 0.199), and *RFC1* (*p* = 0.658).

Our univariate logistic regression found that *ERCC1* was negatively associated with a chemotherapeutic response [restricted cubic spline (RCS) knot 1: OR −0.27, CI −0.504 to −0.032; *p* = 0.036; knot 2: OR = 0.39, CI −0.043 to 0.831; *p* = 0.036], as shown in Figure 4, while none of the other gene candidates exhibited a significant association. With multivariable logistic regression, including age and sex, this association was no longer significant for *ERCC1* (RCS knot 1: OR −0.26, CI −0.508 to −0.003; *p* = 0.062; knot 2: OR 0.37, CI −0.103 to 0.840; *p* = 0.142), nor were any of the other gene candidates significant.

#### 3.3.3. Prediction of Multiple Factors in Predicting Chemotherapeutic Response

Our multivariable logistic regression, which included the age, sex, histological sub-type, and gene candidates found that age, *ABCC3*, *ERCC1*, *RFC1*, and chondroblastic and osteoblastic histology contributed significantly to predicting a response to chemotherapy. The age and gene candidates were modelled as three-knot restricted cubic splines (Table 5).

## 4. Discussion

This is the first prospective African study interrogating the gene expression profiles associated with the chemo-response of conventional osteosarcoma (COS). We analysed relative gene expression data using RT–PCR and the 2^−∆∆CT^ method [18]. We observed a unique pattern of significant downregulation in the mRNA expressions of genes *ABCB1* (*p-glycoprotein*), *ABCC1*, *ERCC1*, *p-53*, and *RFC1* of the COS patients compared to the phenotypically healthy donor muscles. Furthermore, the tumour tissues with histologically confirmed COS exhibited a significant downregulation in their mRNA expression profiles for *ABCB1* (*p-glycoprotein*) and *ABCC3* compared to the healthy donor muscles, as measured using real-time PCR. Although our cohort had variability in their mRNA gene expression, we caution about interpreting our results as the sample size was small, aside from the intrinsic and extrinsic factors that may influence gene expression. Nonetheless, in this cohort, the downregulation in the mRNA expression of *ABCB1* (*p-glycoprotein*), *ABCC1*, *ERCC1*, *p-53*, and *RFC1* in the COS patients suggests the involvement of unknown biological risk factors of genomic instability such as cancer cell mutations, tumour microenvironment (cell types, signalling molecules, and extracellular matrix), epigenetic changes (DNA methylation and histone modification), dysregulation signalling pathways, immune response, and metabolic changes. In 2023, a systematic review described the gene expression patterns of COS as highly variable and heterogeneous. Furthermore, 473 differentially expressed genes (DEGs) were associated with chemotherapy response, and 57 genes were associated with MDR [12]. Unfortunately, not all 57 chemoresistance genes previously identified by the authors could be explored in this series amid our limited resources. The slight increase in COS among Africans suggests unknown risk factors and genetic alterations [19]. The genetic and epigenetic pattern needs to be studied in greater detail amongst individuals of African ethnicity. Our study indicates there are differential expression patterns amongst healthy individuals and COS patients. Novel gene expression profiles may underpin the tumourigenesis of osteosarcoma in African ethnicities; hence, the aggressive nature with an advanced Enneking stage, early metastasis, and impoverished survivals [19,20,21]. Therefore, wide genomic exploratory studies are needed to expand our understanding of this disease.

The *ERCC1* gene is representative of the nucleotide excision repair genes, which assist osteosarcoma tumour cells in repairing the DNA damage caused by chemotherapy drugs, including cisplatin and cyclophosphamide [22]. In our study, the *ERCC1* gene was downregulated in the COS patients, which may suggest it has undergone biological alterations. The dysregulated *ERCC1* gene in the amputated osteosarcoma tumours suggests its participation in the tumourigenesis, aggressive phenotype, and poor chemo-response [23]. In our univariate and multivariate analyses of osteosarcomas, the *ERCC1* gene in the COS tumour specimens was negatively associated with the chemo-response. Contrasting evidence co-exists regarding the associations between *ERCC1* gene expression studies, polymorphisms, and chemo-response in COS [22,23,24,25,26,27]. In 2023, Trujillo-Paolillo et al. and colleagues concurred with our findings that *ERCC1* gene expression was negatively associated with chemo-response [23]. However, Nathrath et al. (2012) found no correlation between the *ERCC1* gene and chemo-response in 45 osteosarcoma patients [24]. Meanwhile, Hao et al. (2012) and Zhang et al. (2015) implicated the *ERCC1* single nucleotide polymorphism (SNP) rs11615 in patients with a good chemo-response and prognosis [15,25]. Similarly, a meta-analysis by Liu et al. (2017) concurred with the previous authors and further suggested that the *ERCC1 rs11615* SNP could be a useful genetic marker for predicting osteosarcoma prognosis [25]. These findings contradict Yang et al. (2012) and Li et al. (2014), who did not find a significant association between the *ERCC1* polymorphism and chemo-response [26,27]. Fanelli et al. (2020) used the in vitro validation of candidate DNA repair-related therapeutic targets and drugs for tailored treatment in cisplatin-resistant osteosarcoma and found the *ERCC1* gene as one of the main therapeutic targets [28]. Therefore, *ERCC1* mRNA expression and polymorphisms warrant further investigations in osteosarcoma patients.

On the other hand, the *ABC* transporters (including the *ABCC3* and *ABCB1* gene products) are transmembrane ATP-dependent efflux pumps that block chemotherapy drugs from entering osteosarcoma tumour cells, leading to poor outcomes and systemic toxicity [9,12,13]. In our study, the *ABCC3* and *ABCB1* genes in the osteosarcoma tissues were significantly downregulated compared to the healthy donors. This may suggest that they are mutated, participate in tumour suppression, or are influenced by cancer cells’ epigenetic patterns, which can silence the gene that will normally suppress the tumour. In our multivariate logistic regression, *ABCC3* was a negative predictor of chemo-response. In 2021, Ramírez-Cosmes et al. implicated the *ABCC3* gene in chemotherapeutic responses in various cancers including lung, colon, breast, bladder, and gliomas [29]. Furthermore, in 2022, a systematic appraisal by Hurkmans et al., which focused on the role of genetic variants in COS, found *ABCC3* and *ABCB1* variants conferred advantages in chemotherapeutic responses, relapse, and event-free and overall survival [30]. Baldini et al. found that *ABCB1* (*P-pg*) is associated with MDR of osteosarcoma, specifically referencing conventional chemotherapy drugs, including doxorubicin [31]. Furthermore, with respect to *P-pg*, overexpression is associated with cisplatin efficacy in osteosarcoma patients [13]. The expression of *ABCB1* and *ABCC3* in osteosarcoma metastatic tumour biopsies confers impoverished event-free and overall survival outcomes [23].

In contrast to drug efflux pumps, the reduced folate carrier 1 (*RFC1*) gene is responsible for the intracellular transportation of chemotherapy drugs, including high-dose methotrexate (HDMTX), and its decreased expression is thought to lead to a poor chemotherapeutic response [9,14,32,33]. On the other hand, in 2022, Wu et al. conducted an integrative analysis of the expression of the *RFC* family of genes, which exhibited an increase in sarcoma tissues [34]. In our osteosarcomas, the expression of *RFC1* negatively predicted the chemotherapeutic response. HDMTX is a mainstay therapeutic agent in osteosarcoma; therefore, investigating genetic variants with the HDMTX pathway may provide important insights for future clinical practice [9,14,32,33]. The significant correlation between low *RFC1* mRNA expression at diagnosis, poor histological response, and osteosarcoma recurrence warrants larger-scale investigation [32]. In addition, *p53*, the oncogene, is considered the prototypic tumour suppressor gene in osteosarcoma in that a complete loss of function is required before tumourigenesis [15,35]. Dysregulated or mutant *P53* influences chemotherapeutic responses and impoverished COS survival [36]. In our case, the *p53* gene was downregulated in the COS patients compared to the healthy donors, which may also suggest it has undergone biological alterations, as mentioned previously, which underscores the need for further elucidation.

In 2011, Kubista et al. used differentially expressed genes to classify the histological subtypes of COS and found the genes *prohibitin*, *Annexin 1*, *Annexin 4*, and *GGH* to be involved in chemoresistance [37]. Their bioinformatic analysis showed that the previously mentioned genes were expressed in osteoblastic and non-osteoblastic osteosarcoma subtypes [37]. In our series, the proportions of COS histological subtypes comprised 46% osteoblastic, 32% chondroblastic, 11% mixed lesions, and 11% fibroblastic tumours. Age, osteoblastic subtypes, and chondroblastic subtypes were negative predictors of chemo-response. In contrast, fibroblastic and mixed subtypes were positively predictive of chemo-response. Young age in osteosarcoma confers a better chemo-response, as well as event-free and overall disease survival [3,38]; this is in contrast to increased age > 22, which increases the risk of mortality in COS patients [3]. However, this remains to be seen in our prospective follow-up of this cohort. In 2002, Hauben et al. found a significant proportion of good chemo-responders in the fibroblastic subtype compared to the chondroblastic subtype [39]. Contrary to Simeland et al. (2019), who conducted the largest EURAMOS -1 osteosarcoma clinical trial and found that the histological subtype telangiectatic exhibited better survival than the unspecified conventional subtypes [5]. Good chemotherapy response in COS confers better survival [5,6,7], but unfortunately, 84% of our patients exhibited chemoresistance. This may suggest unique genetic expression in our African population.

The strengths of this study include its prospective design, and that it is the first study of this type in the African population to investigate the topic. The sampling strategy used stringent inclusion criteria for only conventional subtypes of osteosarcoma and exclusion criteria for other variants. In addition, the genes investigated underpinned COS tumourigenesis, MAP pharmacokinetics, pharmacodynamics, and MDR mechanisms. Owing to the rarity of osteosarcoma, there is a selection bias associated with the small sample size, and the lack of data regarding mRNA expression measurements in the African population, which prevented us from conducting a formal power calculation for the study. This restriction compromises the statistical power and generalizability of the findings, making it difficult to draw robust conclusions about gene expression profiles and their clinical relevance. Three patients were not available for the final analysis, of which two of those abandoned chemotherapy and surgery and went to seek alternative traditional medicine, and one demised amid chemotherapy toxicity. Furthermore, there is a selection bias when employing candidate gene selection or pre-selected target analysis rather than an accurate representation of the genome-wide expression profile. Discordances of evidence exist in the standardised experimental protocols, and the use of different technologies for data acquisition and analysis when investigating osteosarcoma gene expression, make it challenging to compare the results [8,9,10,11,12]. We used RT–PCR and the 2^−∆∆CT^ method, which may introduce another selection bias [18].

## 5. Conclusions

In this South African population, we observed a significant downregulation in the mRNA expression of genes *ABCB1*, *ABCC3*, *ERCC1*, *RFC1*, and *p53* in the conventional osteosarcoma (COS) patients compared to the healthy individuals. Furthermore, compared to the healthy individuals, the tumour tissues with COS exhibited a significant downregulation in the mRNA expression profiles for *ABCB1* (*p-glycoprotein*) and *ABCC3.* Our univariate logistic regression analysis revealed several predictors of chemotherapeutic response, including *ERCC1*, along with osteoblastic, fibroblastic, and mixed tumour subtypes. The multivariable analysis further underscored the significance of age, *ABCC3*, *ERCC1*, *RFC1*, and specific tumour histologies in predicting treatment outcomes. These may serve as prognostication and therapeutic targets. Larger-scale research is needed on the African continent.

## Figures and Tables

**Figure 1 cancers-16-03240-f001:**
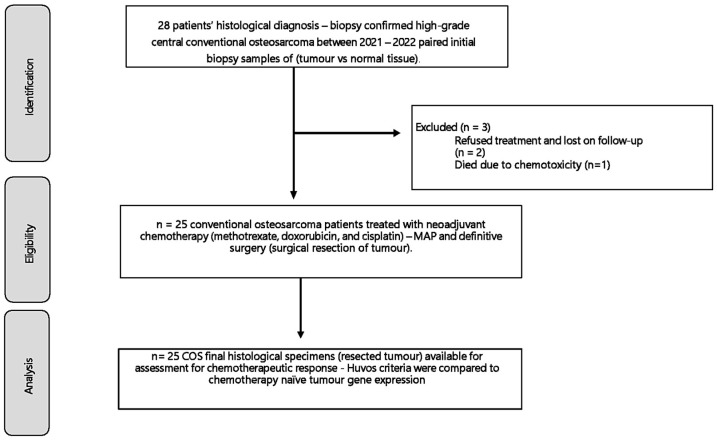
The 28 patients included had a histological biopsy-confirmed high-grade conventional osteosarcoma of the appendicular skeleton. Between 2021 and 2022, after ethical approval, clinical and histological data, and a biopsy of the tumour and normal muscle tissue were collected for a gene expression analysis that was compared to their chemotherapy response.

**Figure 2 cancers-16-03240-f002:**
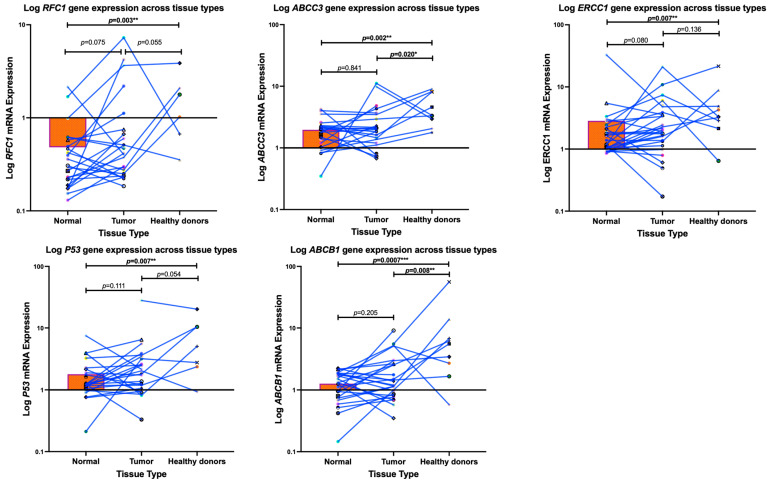
A Mann–Whitney Test was employed to measure the gene expression profiles of conventional osteosarcoma patients (tumours vs. normal muscle tissue) compared to healthy donors (normal muscle) within South Africa. Each dot, shape, and colour represents each patient. Notably, a significant downregulation in the mRNA expressions of genes ABCB1 (p-glycoprotein), ABCC1, ERCC1, p-53, and RFC1 occurred in the conventional osteosarcoma patients compared to the healthy individuals. Furthermore, the tumour tissues of the conventional osteosarcoma patients exhibited a significant downregulation in the mRNA expression of ABCB1 (p-glycoprotein) and ABCC3, compared to healthy patients’ muscles, as measured using real-time PCR. The * *p* < 0.05, ** *p* < 0.01, *** *p* < 0.001 denotes significance.

**Figure 3 cancers-16-03240-f003:**
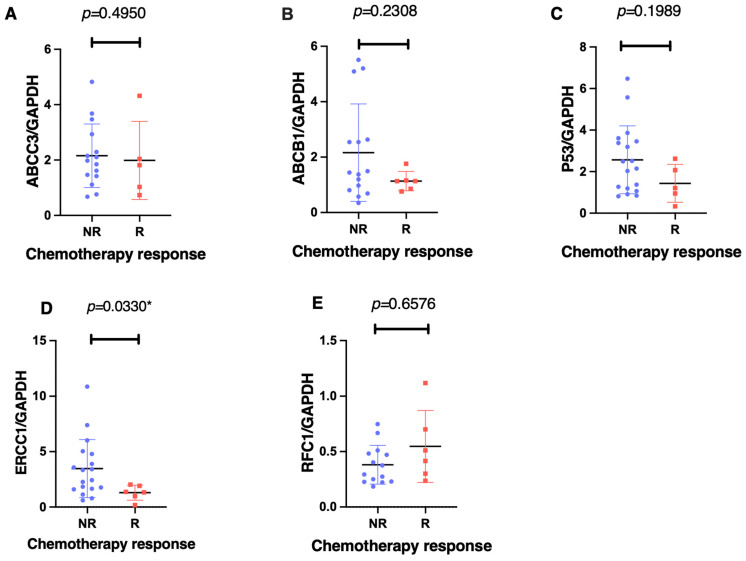
The gene expression associations of ABCC3 (**A**), ABCB1 (**B**), p53 (**C**), ERCC1 (**D**), and RFC1 (**E**), with the effect of the chemotherapy response, as measured using an unpaired, nonparametric, Mann–Whitney, two-tailed experimental design with a significant difference set at *p* < 0.005. Each patient is represented by either a different colour or shape. Notably, there was a significant association between the ERCC1 expression levels and the chemotherapeutic response in the non-responders (N < 90% tumour necrosis—Huvos criteria) as compared to the responders (N > 90% tumour necrosis) post-neoadjuvant chemotherapy. The * *p* < 0.05 denotes significance.

**Figure 4 cancers-16-03240-f004:**
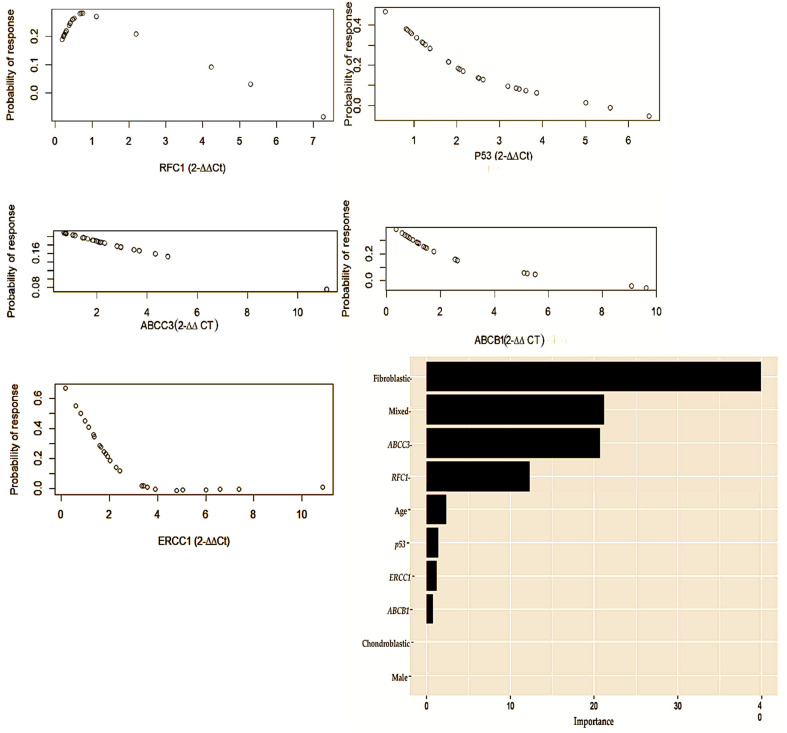
Th gene concentration levels measured using the delta-delta (2^−∆∆CT^) method comprising RFC1, p53, ABCC3, ABCB1, and ERCC1. This multivariate logistic regression represents both the clinical and genetic factors that are predictive of chemotherapeutic responses in osteosarcoma. In accordance with the level of importance, the following variables are presented: age [Restricted cubic spline (RCS) knot: OR −2.5; CI −3.616 to −1.378; *p* = 0.022]; ABCC3 [RCS knot: OR 0.67; CI 0.407 to 0.936, *p* = 0.016]; ERCC1 [RCS knot: OR 0.57; CI 0.235 to 0.901; *p* = 0.044]; RFC1 [RCS knot: OR −1.04; CI −1.592 to −0.487; *p* = 0.035]; chondroblastic [OR −0.83; CI −1.106 to −0.520; *p* = 0.012]; and osteoblastic [OR −1.28; CI −1.664 to −0.91; *p* = 0.007].

**Table 1 cancers-16-03240-t001:** Clinical characteristics of individual patients.

Patient	Age	Sex	Tumour Site	MRI Tumour Volume (cm^3^)	ALP* (U/L)	LDH* (U/L)	Metastasis	Enneking Stage	Histological Subtypes	Surgery Performed	Neoadjuvant Chemotherapy	Chemotherapy Response (%) Huvos Grade
1	21	M	Femur, distal	79,534	144	256	Yes	III	Fibroblastic	Yes	Yes	90 (Huvos III)
2	12	F	Femur, distal	113,256	592	736	Yes	III	Osteoblastic	Yes	Yes	75 (Huvos II)
3	31	F	Femur, distal	160,025	249	624	No	IIB	Osteoblastic	Yes	Yes	50 (Huvos II)
4	17	M	Femur, proximal	58,578	604	626	No	IIB	Osteoblastic	Yes	Yes	75 (Huvos II)
5	16	M	Pelvis	237,186	837	1250	No	IIB	Osteoblastic	No	No	N/A
6	6	M	Femur, proximal	177,907	181	1972	Yes	III	Chondroblastic	Yes	Yes	20 (Huvos I)
7	17	M	Radius, distal	9019	266	432	No	IIB	Chondroblastic	Yes	Yes	10 (Huvos I)
8	20	F	Femur, distal	6029	46	112	No	IIB	Chondroblastic	Yes	Yes	15 (Huvos I)
9	31	F	Femur, distal	160,024	249	624	No	IIB	Osteoblastic	Yes	Yes	70 (Huvos II)
10	12	F	Humerus, proximal	73,912	723	840	Yes	III	Osteoblastic	No	No	N/A
11	11	M	Femur, distal	53,248	172	93	No	IIB	Mixed	Yes	Yes	85 (Huvos II)
12	27	M	Tibia, proximal	80,757	156	220	No	IIB	Osteoblastic	Yes	Yes	70 (Huvos II)
13	6	M	Femur, distal	168,948	369	663	Yes	III	Chondroblastic	Yes	Yes	25 (Huvos I)
14	11	M	Femur, distal	55,929	257	275	No	IIB	Mixed	Yes	Yes	90 (Huvos III)
15	11	M	Femur, distal	11,719	237	330	No	IIB	Osteoblastic	Yes	Yes	30 (Huvos I)
16	16	M	Tibia, proximal	40,069	140	628	Yes	III	Chondroblastic	Yes	Yes	15 (Huvos I)
17	23	M	Tibia, proximal	107,314	220	642	Yes	III	Osteoblastic	Yes	Yes	60 (Huvos II)
18	16	M	Femur, distal	142,272	369	663	Yes	III	Chondroblastic	Yes	Yes	25 (Huvos I)
19	22	F	Femur, distal	87,107	127	190	No	IIB	Mixed	Yes	Yes	95 (Huvos III)
20	16	M	Femur, distal	50,173	452	2437	Yes	III	Fibroblastic	Yes	Yes	90 (Huvos III)
21	17	M	Femur, proximal	58,578	604	626	No	IIB	Osteoblastic	Yes	Yes	75 (Huvos II)
22	6	M	Pelvis	76,452	103	112	No	IIB	Chondroblastic	No	No	N/A
23	14	F	Femur, distal	79,534	144	258	Yes	III	Fibroblastic	Yes	Yes	90 (Huvos III)
24	22	M	Femur, distal	87,107	127	190	No	IIB	Osteoblastic	Yes	Yes	85 (Huvos II)
25	14	F	Femur, distal	50,375	237	330	Yes	III	Osteoblastic	Yes	Yes	75 (Huvos II)
26	11	F	Femur, distal	70,152	98	129	No	IIB	Chondroblastic	Yes	Yes	5 (Huvos I)
27	12	F	Tibia, proximal	113,256	140	628	Yes	III	Osteoblastic	Yes	Yes	70 (Huvos II)
28	16	M	Tibia, proximal	50,273	592	736	Yes	III	Chondroblastic	Yes	Yes	15 (Huvos I)

ALP* (alkaline phosphatase). LDH* (lactate dehydrogenase).

**Table 2 cancers-16-03240-t002:** A cDNA synthesis reaction is set up for a single reaction.

Master Mix	x1 Reaction Volume
5× VILO™ Reaction Mix	4 μL
10× SuperScript™ Enzyme Mix	2 μL
RNA (up to 25 μg)	X μL
DEPC-treated water	to 20 μL

**Table 3 cancers-16-03240-t003:** Forward and reverse primer sequences used in RT-PCR reaction.

Gene	Forward Sequence	Reverse Sequence
*ABCC3*	5′-TGGGGTGAAGTTTCGTACTGG-3′	reverse 5′-CACGTTTGACTGAGTTGGTGATA-3′
*ABCB1*	5′-TTGCTGCTTACATTCAGGTTTCA-3′	5′-AGCCTATCTCCTGTCGCATTA-3′
*ERCC1*	5′-CCTTATTCCGATCTACACAGAGC-3′	5′-TATTCGGCGTAGGTCTGAGGG-3′
*RFC1*	5′-TTGAACGAGATGAGGCCAAGT-3′	5′-CCCTTTCTTGCGGAGATTCTCT-3′
*p53*	5′-ACAGCTTTGAGGTGCGTGTTT-3′	5′-CCCTTTCTTGCGGAGATTCTCT-3′
*GAPDH*	5′-TCCACCACCCTGTTGCTGTA-3′	5′-ACCACAGTCCATGCCATCAC-3′

**Table 4 cancers-16-03240-t004:** The descriptive statistics results of the cohort.

Patients	Summary Measure ^i^ (*n* = 28)
Median age (years)	16 (IQR 11.3–20.8)
Sex	
Male	18 (64.3%)
Female	10 (35.7%)
Tumour location	
Femur, distal	16 (57%)
Tibia, proximal	5 (18%)
Other sites	7 (25%)
Metastasis at diagnosis	
Yes	13 (46%)
No	15 (54%)
Histological subtype	
Osteoblastic	13 (46%)
Chondroblastic	9 (32%)
Fibroblastic	3 (11%)
Mixed	3 (11%)
Median MRI tumour volume (cm^3^)	7799 cm^3^ (IQR 5109–1133; CI = 664 to 1092)
Median alkaline phosphatase (ALP = U/L)	2370 U/L (IQR 1410–431; CI = 2184 to 3841)
Median lactate dehydrogenase (LDH = U/L)	624 U/L (IQR 2290–6630; CI = 3861 to 8012)
Definitive surgery performed	
Yes	25 (89%)
No	3 (11%)
Neoadjuvant chemotherapy	
Yes	25 (89%)
No	3 (11%)
Chemotherapy response (tumour necrosis)	
Non-responder (NR < 90%)	21 (84%)
Responder (R = or >90%)	4 (16%)
Median follow time in months	12.7 (IQR 9–17)
Demised during follow up	3 (11%)

(i) Frequency (Percentage), unless stated otherwise.

**Table 5 cancers-16-03240-t005:** Multivariate logistic regression predicting chemotherapeutic response.

Patient Characteristics and Gene Candidates	Odds Ratio (95% Confidence Interval)	*p* Value
Age	−0.11 (−0.171 to −0.039)	0.052
Age `	−2.5 (−3.616 to −1.378)	0.022 *
Male sex	−0.17 (−0.369 to −0.021)	0.179
ABCC3	0.67 (0.407 to 0.936)	0.016 *
ABCC3 `	−1.02 (−1.463 to −0.575)	0.020 *
ABCB1	−0.20 (−0.467 to 0.063)	0.232
ABCB1 `	0.52 (−0.322 to 1.355)	0.314
p53	−0.02 (−0.293 to 0.244)	0.869
p53 `	0.23 (0.014 to 0.446)	0.128
ERCC1	−0.37 (−0.599 to −0.133)	0.054
ERCC1 `	0.57 (0.235 to 0.901)	0.044 *
RFC1	−1.04 (−1.592 to −0.487)	0.035 *
RFC1 `	1.43 (0.647 to 2.217)	0.037 *
Osteoblastic	−1.28 (−1.664 to −0.901)	0.007 *
Chondroblastic	−0.81 (−1.106 to −0.520)	0.012 *
Fibroblastic	0.11 (−0.199 to 0.418)	0.536
Mixed	−0.35 (−0.602 to −0.121)	0.374

Multivariable logistic regression, which included age, sex, histological sub-type, and gene candidates, found age, *ABCC3*, *ERCC1*, *RFC1*, chondroblastic histology, and osteoblastic histology were predictors of response to chemotherapy (Table 5) AIC −38.748. Age and gene candidates are modelled as three-knot restricted cubic splines. * *p* < 0.05; ` first knot.

## Data Availability

The authors confirm that the data supporting the findings of this study are available within the article.

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
