# Peer review of "Unique Gene Expression Profiles within South Africa Are Associated with Varied Chemotherapeutic Responses in Conventional Osteosarcoma"

_cancers, 2024, doi:10.3390/cancers16183240_

Round 1
Reviewer 1 Report (Previous Reviewer 1)
Comments and Suggestions for Authors
My primary concern remains the number of osteosarcoma patients.
Despite the addition of 9 healthy donor samples for comparative purposes, the failure to augment the osteosarcoma patient cohort means the study still suffers from a limited sample size. This restriction compromises the statistical power and generalizability of the findings, making it difficult to draw robust conclusions about gene expression profiles and their clinical relevance. Without a larger patient cohort, the study’s findings may be affected by high variability and may not be applicable to a broader patient population.
Comments on the Quality of English Languageminor editing
Author Response
Reviewer 1
|
|
Can be improved |
Must be improved |
Not applicable |
|
|
Does the introduction provide sufficient background and include all relevant references? |
(x) |
( ) |
( ) |
( ) |
|
Is the research design appropriate? |
( ) |
( ) |
(x) |
( ) |
|
Are the methods adequately described? |
(x) |
( ) |
( ) |
( ) |
|
Are the results clearly presented? |
( ) |
( ) |
(x) |
( ) |
|
Are the conclusions supported by the results? |
( ) |
( ) |
(x) |
( ) |
Comments and Suggestions for Authors
My primary concern remains the number of osteosarcoma patients.
Despite the addition of 9 healthy donor samples for comparative purposes, the failure to augment the osteosarcoma patient cohort means the study still suffers from a limited sample size. This restriction compromises the statistical power and generalizability of the findings, making it difficult to draw robust conclusions about gene expression profiles and their clinical relevance. Without a larger patient cohort, the study’s findings may be affected by high variability and may not be applicable to a broader patient population.
Response
We wish to highlight that there is currently a lack of data regarding mRNA expression measurements in the African population, which has prevented us from conducting a formal power calculation for our study. This work serves as a pilot study and lays the groundwork for future research to investigate the role of gene expression in disease within this demographic. Our findings will be valuable for subsequent studies as a basis for power calculations, marking it the first of its kind in this context. Unfortunately, due to the absence of comparable studies, we are unable to perform a power calculation at this time.
Reviewer 2 Report (Previous Reviewer 2)
Comments and Suggestions for Authors
The resubmission shows significant improvement and now appears satisfactory. However, there are still several comments that need to be addressed before it can be considered for publication.
1.There are still minor grammatical and typographical errors scattered throughout the document, suggesting that a thorough proofreading has not yet been completed, such as on Line 146 where 'only' is awkwardly placed and should be revised to 'The study proceeded once all the necessary ethical and local regulatory approvals were obtained', Line 222 where 'Reported as' should be 'It was reported as', Line 275 where 'suggest' should be 'suggests', Line 164 where the phrase 'the five candidate genes selection' is grammatically incorrect, Line 294 where 'With' should be 'Using', and Line 164 where 'heterogenous' should be 'heterogeneous'.
2.Figures such as Figure 2 and others are present but need clearer reorganization, as some content (like text and figures) still appears small and less readable.
3.Figure 2 lacks mean values and error bars; please include them. Additionally, the authors have used shapes and colors to distinguish samples from different sources. It is recommended to connect data points from the same patient or sample source with lines to improve clarity and visualization.
4.The paper could benefit from a more explicit discussion of the study’s limitations, particularly the small sample size. Acknowledging this and discussing how future studies could address it.
Comments on the Quality of English LanguageModerate editing of English language required
Author Response
Reviewer 2
|
|
Can be improved |
Must be improved |
Not applicable |
|
|
Does the introduction provide sufficient background and include all relevant references? |
( ) |
(x) |
( ) |
( ) |
|
Is the research design appropriate? |
( ) |
(x) |
( ) |
( ) |
|
Are the methods adequately described? |
( ) |
(x) |
( ) |
( ) |
|
Are the results clearly presented? |
( ) |
(x) |
( ) |
( ) |
|
Are the conclusions supported by the results? |
( ) |
(x) |
( ) |
( ) |
Comments and Suggestions for Authors
The resubmission shows significant improvement and now appears satisfactory. However, there are still several comments that need to be addressed before it can be considered for publication.
1.There are still minor grammatical and typographical errors scattered throughout the document, suggesting that a thorough proofreading has not yet been completed, such as on Line 146 where 'only' is awkwardly placed and should be revised to 'The study proceeded once all the necessary ethical and local regulatory approvals were obtained', Line 222 where 'Reported as' should be 'It was reported as', Line 275 where 'suggest' should be 'suggests', Line 164 where the phrase 'the five candidate genes selection' is grammatically incorrect, Line 294 where 'With' should be 'Using', and Line 164 where 'heterogenous' should be 'heterogeneous'.
2.Figures such as Figure 2 and others are present but need clearer reorganization, as some content (like text and figures) still appears small and less readable.
3.Figure 2 lacks mean values and error bars; please include them. Additionally, the authors have used shapes and colors to distinguish samples from different sources. It is recommended to connect data points from the same patient or sample source with lines to improve clarity and visualization.
4.The paper could benefit from a more explicit discussion of the study’s limitations, particularly the small sample size. Acknowledging this and discussing how future studies could address it.
Response
Line 146 – “only” removed.
Line 164 – grammatical errors revised to “heterogeneous”
Line 222 – revised to “it was reported as”
Line 275 – revised to “suggests”
Line 294 – revised to “using”
Figures 2 - enlarged, reorganized to make it clear including text content. We have included mean values and error bars, and connection of data points from same patient.
Line 503 – 508 – study limitations: We wish to highlight that there is currently a lack of data regarding mRNA expression measurements in the African population, which has prevented us from conducting a formal power calculation for our study. This work serves as a pilot study and lays the groundwork for future research to investigate the role of gene expression in disease within this demographic. Our findings will be valuable for subsequent studies as a basis for power calculations, marking it the first of its kind in this context. Unfortunately, due to the absence of comparable studies, we are unable to perform a power calculation at this time.
Reviewer 3 Report (New Reviewer)
Comments and Suggestions for Authors
Dear Author
The manuscript can be published after solving following issues
1- The correct spelling “cDNA” should takes the place of “CDNA” in all parts
2- Why some parts are underline?
3- Several publications indicated to the role of ABCC3, ERCC1, and RFC1 in drug resistance. The novelty of work is the target population? Please consider the following publications in the discussion part and add to reference list.
Ramírez-Cosmes A, Reyes-Jiménez E, Zertuche-Martínez C, Hernández-Hernández CA, García-Román R, Romero-Díaz RI, Manuel-Martínez AE, Elizarrarás-Rivas J, Vásquez-Garzón VR. The implications of ABCC3 in cancer drug resistance: can we use it as a therapeutic target?. American Journal of Cancer Research. 2021;11(9):4127.
Fanelli M, Tavanti E, Patrizio MP, Vella S, Fernandez-Ramos A, Magagnoli F, Luppi S, Hattinger CM, Serra M. Cisplatin resistance in osteosarcoma: in vitro validation of candidate DNA repair-related therapeutic targets and drugs for tailored treatments. Frontiers in oncology. 2020 Mar 10;10:331.
Marchandet L, Lallier M, Charrier C, Baud’huin M, Ory B, Lamoureux F. Mechanisms of resistance to conventional therapies for osteosarcoma. Cancers. 2021 Feb 8;13(4):683.
Hurkmans EG, Brand AC, Verdonschot JA, Te Loo DM, Coenen MJ. Pharmacogenetics of chemotherapy treatment response and-toxicities in patients with osteosarcoma: a systematic review. BMC cancer. 2022 Dec 19;22(1):1326.
Wu G, Zhou J, Zhu X, Tang X, Liu J, Zhou Q, Chen Z, Liu T, Wang W, Xiao X, Wu T. Integrative analysis of expression, prognostic significance and immune infiltration of RFC family genes in human sarcoma. Aging (Albany NY). 2022 Apr 4;14(8):3705.
Comments on the Quality of English LanguageMinor editing of English language required.
Author Response
|
Yes |
Can be improved |
Must be improved |
Not applicable |
|
|
Does the introduction provide sufficient background and include all relevant references? |
( ) |
(x) |
( ) |
( ) |
|
Is the research design appropriate? |
(x) |
( ) |
( ) |
( ) |
|
Are the methods adequately described? |
(x) |
( ) |
( ) |
( ) |
|
Are the results clearly presented? |
(x) |
( ) |
( ) |
( ) |
|
Are the conclusions supported by the results? |
(x) |
( ) |
( ) |
( ) |
Comments and Suggestions for Authors
Dear Author
The manuscript can be published after solving the following issues
1- The correct spelling of “cDNA” should take the place of “CDNA” in all parts
2- Why are some parts underlined?
3- Several publications indicated to the role of ABCC3, ERCC1, and RFC1 in drug resistance. The novelty of the work is the target population. Please consider the following publications in the discussion part and add them to the reference list.
Ramírez-Cosmes A, Reyes-Jiménez E, Zertuche-Martínez C, Hernández-Hernández CA, García-Román R, Romero-Díaz RI, Manuel-Martínez AE, Elizarrarás-Rivas J, Vásquez-Garzón VR. The implications of ABCC3 in cancer drug resistance: can we use it as a therapeutic target? American Journal of Cancer Research. 2021;11(9):4127.
Fanelli M, Tavanti E, Patrizio MP, Vella S, Fernandez-Ramos A, Magagnoli F, Luppi S, Hattinger CM, Serra M. Cisplatin resistance in osteosarcoma: in vitro validation of candidate DNA repair-related therapeutic targets and drugs for tailored treatments. Frontiers in oncology. 2020 Mar 10; 10:331.
Marchandet L, Lallier M, Charrier C, Baud’huin M, Ory B, Lamoureux F. Mechanisms of resistance to conventional therapies for osteosarcoma. Cancers. 2021 Feb 8;13(4):683.
Hurkmans EG, Brand AC, Verdonschot JA, Te Loo DM, Coenen MJ. Pharmacogenetics of chemotherapy treatment response and toxicities in patients with osteosarcoma: a systematic review. BMC cancer. 2022 Dec 19;22(1):1326.
Wu G, Zhou J, Zhu X, Tang X, Liu J, Zhou Q, Chen Z, Liu T, Wang W, Xiao X, Wu T. Integrative analysis of expression, prognostic significance and immune infiltration of RFC family genes in human sarcoma. Aging (Albany NY). 2022 Apr 4;14(8):3705.
Response
- CDNA corrected to “cDNA”
- The underlined parts are the changes made previously in the manuscript.
- References added on the discussion as underlined and added on the reference section as follows: line 582 – 583, line 630 – 632, line 633 – 635, line 636 – 638, line 647 – 649.
This manuscript is a resubmission of an earlier submission. The following is a list of the peer review reports and author responses from that submission.
Round 1
Reviewer 1 Report
Comments and Suggestions for Authors
This paper analyzes the gene expression profiles associated with conventional osteosarcoma (COS) in an African population and highlights potentially important "trends" in gene expression patterns.
In my opinion, there are some points that require attention:
- The observed correlations between gene expression levels and chemotherapy response were not statistically significant. Although the differences did not reach statistical significance, they may still have biological significance and could potentially influence chemotherapy response. However, this limits the strength of the conclusions drawn from the study.
-The study addresses an important aspect of osteosarcoma research, so the authors should consider expanding the sample size (by expanding the timeframe or collaborating with another team to increase the number of patients) to strengthen the impact of the research. The sample size of 28 patients for a prospective study investigating gene expression profiles in COS appears relatively small. Typically, larger sample sizes are desirable to enhance statistical power and generalizability of findings. Of course, the adequacy of the sample size depends on various factors, including the rarity of the condition being studied.
-Moreover, the authors should provide a brief explanation of the roles of the genes studied in the introduction. This would help readers who may not be familiar with the specific genes to initially understand their significance in the context of osteosarcoma and chemotherapy response.
-Finally, the authors could more clearly state how many additional studies have investigated these genes for osteosarcoma, in what ethnicities, and in which other types of cancer they have been examined and correlated.
Comments on the Quality of English Language
minor editing
Author Response
Response 1 :
Conclusions changed - Line 449 – 459 (manuscript).
Response 2:
Our systematic review on the topic informed the gene selection – published last year, 2023, in Genes, MDPI journal. Reference: Mthethwa PG, Marais LC, Ramsuran V, Aldous CM. A Systematic Review of the Heterogenous Gene Expression Patterns Associated with Multidrug Chemoresistance in Conventional Osteosarcoma. Genes. 2023;14(4):832.
Previous studies discussed the small systematic review sample sizes; hence, international collaborative research has been established. However, this kind of collaborative work has excluded patients of African descent (Mthethwa et al., 2023, Systematic Review).
However, we agree that there is selection bias in candidate gene selection or pre-selected target analysis rather than an accurate representation of the genome-wide expression profile. Discordances of evidence in standardised experimental protocols, using different technologies for data acquisition and analysis investigating osteosarcoma gene expression, make it challenging to compare the results. We used RT–PCR and the 2- ∆∆CT method, which may introduce another selection bias.
In addition, owing to the rarity of osteosarcoma, the small sample size and resource-constraint environment also introduce bias.
Large-scale collaborative genomic-wide research is needed as our results require further validation at a broader scale for future prognostication and therapeutic targets in COS in our context.
Response 3:
Paragraph lines 98 – 122 were added to the manuscript.
Response 4:
Reference:
Our systematic review on the topic informed the gene selection – published last year, 2023, in Genes, MDPI journal. Reference: Mthethwa PG, Marais LC, Ramsuran V, Aldous CM. A Systematic Review of the Heterogenous Gene Expression Patterns Associated with Multidrug Chemoresistance in Conventional Osteosarcoma. Genes. 2023;14(4):832.

Reviewer 2 Report
Comments and Suggestions for Authors
In the manuscript entitled "Unique Gene Expression Profiles within South Africa are Associated with Varied Chemotherapeutic Responses in Conventional Osteosarcoma," Phakamani G. Mthethwa et al. delves into how specific gene expressions—namely ABCC3, ABCB1, ERCC1, RFC1, and p53—correlate with the efficacy of chemotherapy treatments in South African patients facing conventional osteosarcomas. Through a meticulous comparison of gene expression in tumor tissues against normal tissues, the research uncovers a notable, albeit non-significant, uptick in the expressions of ERCC1 and RFC1 within tumor samples, pinpointing an increased expression of ERCC1 as a critical marker for poor chemotherapy response. This nuanced understanding of gene expression not only sheds light on the complex dynamics at play in chemotherapy efficacy but also identifies age, histological tumor subtypes (with a focus on chondroblastic and osteoblastic osteosarcomas), and the expressions of the genes ABCC3, ERCC1, and RFC1 as pivotal predictors of treatment outcomes. By offering a detailed exploration of these genetic markers, the study paves the way for refining chemotherapy protocols, aiming for a tailored treatment approach that could significantly enhance the prognosis for osteosarcoma patients in South Africa, thus marking an advancement in the personalized treatment landscape for this challenging disease. The authors have not effectively analyzed and leveraged the valuable clinical resources they collected. Additionally, they failed to provide information on ethical approval, leaving it unclear whether the study adheres to ethical guidelines. There are several comments that could help the author improve their draft research work.
1.It appears that the work has not been approved ethically; therefore, the author should provide related claims and supporting materials.
2.The figure appears to be non-uniform, with some content too small or unclear to read. It should be reorganized to improve both readability and visual appeal.
3.Tables 2 and 3 should be relocated to the supplementary data section.
4.In Figure 2, the legends should be moved from within the figures to the designated figure legend section.
5.There are minor grammatical and typographical errors throughout the document. A thorough proofreading session would help in correcting these mistakes, contributing to the overall quality of the writing.
6.Why did the analysis specifically target the ATP binding cassette subfamily C members (ABCC3 and ABCB1), excision repair cross-complementing group 1 (ERCC1), replication factor C subunit 1 (RFC1), and tumor protein 53 (p53) genes? Given that these genes did not show significant changes, it suggests that they may not be pivotal factors within the patient samples collected. Additionally, some genes function as oncogenes or tumor suppressors through mutations rather than depending on their expression levels.
Comments on the Quality of English LanguageMinor editing of English language required
Author Response
Response 1:
Please see the attachments for Ethical approval before commencing the study.
The consent forms have been forwarded.
Response 2:
The figure’s quality has been improved for readability and visual appeal.
Response 3:
Noted, will do. (see attachment of supplemental material).
Response 4:
Noted, it has been done.
Response 5:
Noted.
Response 5:
Our systematic review on the topic informed the gene selection – published last year, 2023, in Genes, MDPI journal. Reference: Mthethwa PG, Marais LC, Ramsuran V, Aldous CM. A Systematic Review of the Heterogenous Gene Expression Patterns Associated with Multidrug Chemoresistance in Conventional Osteosarcoma. Genes. 2023;14(4):832.
However, we agree that there is selection bias in candidate gene selection or pre-selected target analysis rather than an accurate representation of the genome-wide expression profile. Discordances of evidence in standardised experimental protocols, using different technologies for data acquisition and analysis investigating osteosarcoma gene expression, make it challenging to compare the results. We used RT–PCR and the 2- ∆∆CT method, which may introduce another selection bias.
In addition, owing to the rarity of osteosarcoma, the small sample size and resource-constraint environment also introduce bias.
Large-scale collaborative genomic-wide research is needed as our results require further validation at a broader scale for future prognostication and therapeutic targets in COS in our context.

Round 2
Reviewer 1 Report
Comments and Suggestions for Authors
In my opinion, the study continues to exhibit limitations, with insufficiently robust correlations. Consequently, I maintain the view that it does not meet the standards for acceptance. I suggest that the authors consider expanding either the sample size or the gene panel to strengthen the study's validity and reliability.
Comments on the Quality of English Language
minor editing
Author Response
Dear editorial team and reviewers.
We apologise for the delays in the response to this paper. And this was due to our constraints on resources and ethical approvals. We have improved our methodology by expanding the sample size by including healthy individuals operated for elective orthopaedic procedures (biopsied healthy muscles which would discarded anyways during their normal procedures).
Figure 2 depicts the new changes and provides an analysis of the results. Hence, this manuscript has major revisions in discussion, abstract, and conclusions.
Reviewer 2 Report
Comments and Suggestions for Authors
Thank you for the author's response, which has addressed some of my concerns and comments, resulting in improvements to the revised version. However, several comments still require further attention.
1.The figures remain unclear and lack uniformity. Please ensure each figure is legible and comfortable to read.
2.When you submit your manuscript, could you highlight all the changes you've made? However, I'd recommend not using Word's track changes feature for this. It's really about keeping the document clean and tidy, which is just a basic way to show respect for the work of the editors and reviewers.
3.When responding to reviewer comments, could you provide detailed, point-by-point replies? It's important to clearly explain to the editors and reviewers what changes you've made, how you've made them, and if there are any suggestions you couldn't follow, please offer a well-reasoned explanation. Simply replying with "Noted" doesn't really communicate the depth of your revisions or your engagement with the feedback.
4. It s really a waste of my time to do this work for your draft.
Comments on the Quality of English LanguageMinor editing of English language required
Author Response
Dear editorial and reviewer,
We apologise for the delayed review process as we have to do major revisions and get an updated ethical approval:
The changes made to improve the manuscript comprise:
- Methodology as suggested by one reviewer, expansion of the sample size by including healthy individuals (normal muscle biopsies) to compare with our conventional osteosarcoma patients' mRNA gene expression
- The results are summarised in Figure 2 of the manuscript, which improved the paper results, discussions, conclusions, and abstracts.
- We have improved the uniformity of the figures.
- The Microsoft Word track changes have been unable to.